# Analysis of Salivary Levels of IL-1β, IL17A, OPG and RANK-L in Periodontitis Using the 2017 Classification of Periodontal Diseases—An Exploratory Observational Study

**DOI:** 10.3390/jcm12031003

**Published:** 2023-01-28

**Authors:** Marta Relvas, Ricardo Silvestre, Maria Gonçalves, Cristina Cabral, Ana Mendes-Frias, Luís Monteiro, Alexandra Viana da Costa

**Affiliations:** 1University Institute of Health Sciences (IUCS-CESPU), 4585-116 Gandra, Portugal; 2Oral Pathology and Rehabilitation Research Unit (UNIPRO), (IUCS-CESPU), 4585-116 Gandra, Portugal; 3Life and Health Sciences Research Institute (ICVS), School of Medicine, University of Minho, 4710-057 Braga, Portugal; 4ICVS/3B’s–PT Government Associate Laboratory, 4805-017 Braga/Guimarães, Portugal; 5TOXRUN–Toxicology Research Unit, University Institute of Health Sciences, Advanced Polytechnic and University Cooperative (CESPU), CRL, Rua Central de Gandra, 1317, 4585-116, Gandra, PRD, Portugal

**Keywords:** periodontitis, saliva, IL-1β, RANK-L, OPG, IL17-A, biomarkers, grade, stage

## Abstract

Periodontitis is a chronic disease with a high overall prevalence. It involves a complex interplay between the immune-inflammatory pathways and biofilm changes, leading to periodontal attachment loss. The aims of this study were (i) to assess whether the salivary IL-1β, IL-17A, RANK-L and OPG levels have the potential to discriminate between the mild and severe periodontitis conditions; and (ii) to enable diagnostic/prognostic actions to differentiate between distinct levels of the disease. The analysis of the clinical parameters and the evaluation of the salivary immunomediators levels by means of a multiplex flow assay revealed a statistically significantly higher level of IL-1β in the periodontitis III/IV patients, as well as a higher level of RANK-L in the periodontitis III/IV and I/II patients, when compared to the healthy controls. Furthermore, the grade C periodontitis patients presented a significantly higher level of RANK-L compared to the grade B and grade A patients. In the grade C patients, IL-1β had a positive correlation with the PPD and CAL indices and RANK_L had a positive correlation with CAL. The evidence emerging from this study associates the salivary IL-1β and RANK-L levels with an advanced stage of periodontitis, stage III/IV, and with grade C, suggesting the possible cooperative action of both in the inflammatory and bone loss events. In addition to IL-1β, RANK-L could be considered a combined diagnostic biomarker for periodontitis.

## 1. Introduction

Periodontitis is one of the world’s most common chronic human diseases, and it has a significant impact on oral health [1,2]. It is a biofilm-induced inflammatory disease involving a complex interplay between the immune-inflammatory pathways and symbiotic ecological changes, leading to periodontal attachment loss [3]. As periodontitis manifestations result from prolonged and/or exacerbated host inflammatory reactions up to the subgingival dysbiotic microbiome, it can be speculated that the periodontal levels of pro-inflammatory mediators may be higher in patients with faster rates of progression [4].

Despite the present know-how concerning periodontal diseases’ etiology and therapy [5,6], diagnostic tools are mainly confined to clinical and radiographic findings, which describe former events of the disease (bleeding on probing, probing depth, clinical attachment loss, and radiographic bone loss) [7] and do not always reveal the disease in its most incipient forms. Recently, periodontal research has sought to find effective diagnostic molecular biomarkers to support clinicians in the risk assessment and in the decision-making process, as well as in the improvement of the individual’s therapy. Moreover, the new classification of periodontal diseases has opened up the possibility of using biomarkers as a capital tool for the case definition and classification of periodontal diseases by severity, complexity (stage) and risk of progression (grade), especially in terms of the detection of initial forms of the disease [8].

Saliva is designated as a useful tool for screening and monitoring periodontitis, because it is easily collected [9,10] and reflets the inflammatory status of the whole mouth [11]. Indeed, many cytokines and immunomediators, as detected in saliva, are associated with an inflammatory process developed in unbalanced responses. Bone balance regulation involves an important interplay of soluble mediators, one being the receptor activator of nuclear factor kappa-B ligand (RANK-L) and its physiologic inhibitor osteoprotegerin (OPG). RANK-L, an osteolytic cytokine, binds to RANK (the receptor activator of nuclear factor kappa-B) on the mononuclear cell precursors. After cell differentiation, activated osteoclasts express and secrete RANK, accounting for bone resorptive activity. OPG is a decoy receptor that limits the RANK-L binding to RANK, exerting an osteoprotective effect [12]. The RANKL/OPG ratio has been related to the quantity and severity of periodontal bone destruction [13].

Interleukin-1β (IL-1β) was the first cytokine to be specifically measured in the gingival tissue of patients with chronic periodontitis [14]. Since then, numerous papers have reported the measurement of cytokines in GCF and saliva, confirming the distinct cytokine profile of patients with chronic periodontitis [15,16]. Yet, the results are often discrepant, mainly due to the lack of uniformity in the methodological design of the studies [16]. Finally, IL-17 is a pro-inflammatory cytokine produced by the Th17 subset of CD4+ T-helper cells, and it plays a relevant role in chronic inflammatory responses. Some studies have found evidence that IL-17 and Th17 cells are implicated in periodontitis immunopathogenesis [17].

An “ideal” oral biomarker for periodontitis should manage to diagnose the disease condition and reflect its severity, and it should also monitor the disease treatment and predict its prognosis/progression [18]. Despite previous studies, more evidence is needed to determine the potential of saliva biomarkers in periodontitis, particularly in terms of the timely detection of incipient forms and sites of the disease, as well as their potential usefulness in the new classification of periodontal diseases.

The aims of this study were (i) to assess whether the salivary IL-1β, IL-17A, RANK-L and OPG levels, jointly, have the potential to discriminate between the mild and severe periodontitis site conditions according to the current classification criteria of periodontal diseases; and (ii) to enable diagnostic/prognostic actions to differentiate between the incipient and more advanced levels of the disease.

## 2. Materials and Methods

### 2.1. Study Design and Ethical Approval

This study analyzed patients attending dental clinic appointments at the Clinic Unit of the University Institute of Health Sciences (IUCS-CESPU, Gandra, Portugal) between April 2021 and March 2022. The study was submitted and approved by the ethics commission of the IUCS, with reference CE/IUCS/CESPU-08/21, and it was performed according to the Declaration of Helsinki.

### 2.2. Study Population and Clinical Assessments

#### 2.2.1. Research Participants

The patients were carefully informed via oral and written explanations about the objective and procedures of the study. The patients who agreed to participate in the study were asked to sign an informed consent form and fill out a questionnaire before the periodontal examination. The inclusion criteria were the acceptance of patients between 18 and 70 years old and the presence of at least 18 natural teeth. The exclusion criteria were as follows: pregnant women; subjects with current or a previous history of oral or maxillofacial cancer, radiation or other mucosal pathology, including involving the periodontium; subjects who had undergone periodontal treatment in the past six months; subjects who were currently undertaking oncological treatment or taking bone-related medication; subjects with a medical history of diabetes mellitus or hepatic or renal disease; subjects with autoimmune or autoinflammatory diseases or other severe medical conditions or transmittable diseases; subjects with a history of alcohol or drug abuse; subjects who had taken systemic antimicrobials during the last six months, taken anti-inflammatory medication in the previous four months or routinely used oral antiseptics; and subjects with implants or orthodontic appliances. The recorded data from the anamneses included: gender, age, smoking habits (smoker, ex-smoker [quit smoking less than five years ago] or non-smoker [has not smoked for more than five years]), and oral hygiene habits such as the frequency of toothbrushing and use of dental floss and an interdental brush. The periodontal clinical data included: number of absent teeth; number of teeth with mobility; pocket depth (PPD), as measured as the distance from the gingival free-margin to the bottom of the pocket; gingival recession (REC), as measured as the distance from the enamel–cement junction (CEJ) to the free gingival margin (showing a negative signal whenever the gingival margin is located coronary at the CEJ); clinical attachment loss (CAL); plaque index (PI) [19]; and bleeding on probing (BOP) [20]. These data were registered in six locations per tooth (mesio-vestibular, vestibular, disto-vestibular, mesio-lingual, lingual and disto-lingual) using a CPITN 15 Hu-Friedy Europe Periodontal Probe (Rotterdam, the Netherlands). Full-mouth periapical radiographs were taken with long cone paralleling using Rinn holders. The wisdom teeth were excluded from the analysis.

#### 2.2.2. Case Definition

The periodontally healthy (healthy controls) and periodontitis statuses were defined according to the new consensus of the AAP/EFP (American Academy of Periodontology/European Federation of Periodontology) [8]. The sites from the periodontitis patients were classified as follows: (1) initial to moderate periodontitis sites (mild sites), with CAL and PPD corresponding to those described in periodontitis stages I and II (CAL ≤ 4 mm and PPD ≤ 5 mm); and (2) severe to advanced sites (severe sites), with CAL and PPD corresponding to those described in periodontitis stages III and IV (CAL ≥ 5 mm) [8]. The healthy sites from the healthy subjects (healthy sites) (PPD ≤ 3 mm without BOP) [21] were also obtained.

The grades of periodontitis were estimated considering the evidence or risk of rapid progression in three categories, slow, moderate and rapid progression, respectively indicating Grade A, B and C, and the effects on the patient’s systemic health, as defined by Tonetti et al. [8].

#### 2.2.3. Measurement Reliability and Reproducibility of Examinations

Two senior periodontic specialists (CC, MR) were calibrated according to the measurement criteria using 10 volunteers on 2 different days, 48 h apart. The calibration was achieved via measurements of the same random volunteers by the two examiners, registering the grade of reproducibility. The intra-examiner coefficients of correlation (CCI) were 0.97 and 0.98 for CAL and PD, respectively, and the inter-examiner CCI were 0.98 and 0.98 for CAL and PD, respectively.

#### 2.2.4. Collection of Salivary Samples

Using the spitting method, samples of unstimulated saliva were collected from each patient. The subjects avoided practicing any oral hygiene measure, eating, drinking, and chewing gum for 1 h before the collection of the saliva sample. The samples were stored at −80 °C until further analysis [22].

### 2.3. Quantification of Cytokines by Multiplex Bead-Based Immunoassay Analysis

Prior to the assay, the saliva samples were unfrozen and centrifuged at 6000× *g* for 10 min at 4 °C. The supernatants were recovered, and different aliquots were stored at −80 °C to be posteriorly used. The saliva cytokine levels were measured using a High-Sensitivity Human 4-plex Kit (LegendPlex custom-made panel) (BioLegend, San Diego, CA, USA) multiplex assay using a flow cytometer. The IL-1β, IL-17A, OPG and RANK-L levels were measured, and the assays were performed in V-bottom 96-well plates following the manufacturer’s instructions (25 μL volume/sample).

This custom-made kit consists of a multiplex analysis that quantifies four analytes in the same sample where all the molecules lay in the assay detection range of 2.4 to 10,000 pg/mL (IL-1β 10 ng/mL; IL-17A 10 ng/mL; OPG 100 ng/mL and RANK-L 100 ng/mL reconstitution concentration). The standard curve and the samples were analyzed using an LSR II flow cytometer equipped with FACSDiva software (version 6.1.3) (BD Bioscience, Franklin Lakes, NJ, USA), which could differentiate specific beads for each analyte based on their size and fluorescence intensity, where 4000 events had to be acquired per analyte. The results were analyzed using LEGENDplex v8 software (BioLegend) [23]. The concentration of each analyte was quantified in picograms per milliliter.

### 2.4. Statistical Analysis

The data analysis was performed using the IBM SPSS program (Statistical Program for Social Sciences) version 28.0 for Windows and GraphPad Prism 8 software.

The descriptive statistics were expressed as means and standard deviations for the quantitative variables and as frequencies and percentages for the qualitative variables. The clinical and biochemical data were initially examined for normality with the Shapiro–Wilk test. The normality of the data led to the adoption of a parametric analysis. The chi-square test was used to assess the association between the qualitative variables (gender and smoking habits) among the three groups under study.

An ANOVA was used to compare the periodontal indices and cytokines between the different groups, followed by the Bonferroni test. To analyze the correlation between the clinical parameters and cytokines, Spearman’s correlation coefficient was used. The level of statistical significance used was *p* ≤ 0.05.

## 3. Results

### 3.1. Demographic Characteristics and Periodontal Status

Sixty-eight subjects were selected for the study according to the inclusion and exclusion criteria—that is, 22 periodontally healthy patients (healthy controls), 17 with stage I/II periodontitis and 29 with stage III/IV periodontitis. Of the 46 patients with periodontitis, 7 had grade A periodontitis (15.2%), 13 had grade B periodontitis (28.2%) and 26 had grade C periodontitis (56.6%).

The demographic characteristics of the population studied are shown in Table 1. We observed a statistically significant relation between age and periodontal status (χ^2^ (4) = 28.75; *p* < 0.001), with 20 (55.6%) of the 36 individuals aged over 46 years having stage III/IV periodontitis and 13 (36.1%) having stage I/II periodontitis. Of the 11 individuals aged 25 years or less, only 1 had grade I/II periodontitis.

Regarding the behavioral factors that modify or aggravate the disease, we found a total of 46 (67.6%) non-smokers, 8 (11.8%) ex-smokers and only 14 (20.6%) smokers. Overall, a statistically significant association between smoking habits and periodontal condition was observed ((χ^2^ (4) = 9.54; *p* = 0.049), although it was not associated with the tobacco consumption.

Table 2 presents the periodontal indices of the healthy controls and periodontitis patients. We observed statistically significant differences between the healthy controls and patients with periodontitis I/II and periodontitis III/IV, depending on the periodontal indices. These differences were always established by comparison with the periodontitis III/IV group, which presented higher values for all the periodontal indices.

In all cases, the PI (plaque index) percentage and BOP (bleeding on probing) percentage were higher in the periodontitis patients regardless of staging status than in the healthy controls. We observed that the PI and BOP percentage values were higher in the periodontitis III/IV patients (PI, 65.21 ± 26.01; BOP, 47.05 ± 28.23) compared to the periodontitis I/II patients (PI, 41.82 ± 18.13; BOP, 23.71 ± 14.79) and to the healthy controls (PI, 19.27 ± 12.33; BOP, 6.14 ± 4.89) (*p* < 0.001 for both). Furthermore, statistical differences were noted between the healthy controls (PI, 19.27 ± 12.33; BOP, 6.14 ± 4.89) and periodontitis I/II (PI, 41.82 ± 18.13; BOP, 23.71 ± 14.79) groups (*p* = 0.003; *p* = 0.026). In the analysis of the total PPD (mm) and CAL (mm) parameters, we detected higher values in the periodontitis patients regardless of staging status compared to the healthy controls. In these cases, the periodontitis III/IV patients (PPD, 3.68 ± 1.16 mm; CAL, 4.30 ± 1.27 mm) and periodontitis I/II (PPD, 2.5 ± 0.69; CAL, 2.83 ± 0.86) had higher and significantly different values (*p* < 0.001) compared to the healthy controls (PPD, 1.65 ± 0.25 mm; CAL, 1.67 ± 0.26). Statistical differences were also observed between the healthy controls (PPD, 1.65 ± 0.25 mm; CAL, 1.67 ± 0.26) and periodontitis I/II (PPD, 2.5 ± 0.69; CAL, 2.83 ± 0.86) groups (*p* = 0.008 and *p* = 0.001, respectively).

Moreover, regarding the number of teeth, there were statistically significant differences between the healthy controls and periodontitis III/IV group (*p* < 0.001) and between the periodontitis I/II and periodontitis III/IV groups (*p* < 0.05).

### 3.2. Cellular Items: Analysis of OPG, IL-1β, RANK-L and IL-17A Levels in Saliva

The saliva inflammatory markers were evaluated in all the groups of patients. A significantly higher level of IL-1β in the stage III/IV periodontitis patients (1866.25 ± 1152.15 pg/mL) was observed compared to the healthy controls (913.25 ± 418.20 pg/mL) (*p* = 0.006; Table 3). The periodontitis I/II patients also presented higher levels of IL-1β (1429.38 ± 1037.22 pg/mL) compared to the healthy controls, albeit without statistically significant differences.

More pertinently, we detected statistically significantly higher RANK-L levels in the periodontitis III/IV (49.14 ± 19.55 pg/mL) and periodontitis I/II patients (37.10 ± 10.88 pg/mL) compared to the periodontally healthy controls (28.18 ± 9.68 pg/mL) (*p* < 0.001).

We also observed that the patients with periodontitis III/IV had higher mean values of IL-17A compared to the periodontally healthy controls and the patients with periodontitis I/II, albeit without reaching statistical significance.

Unexpectedly, we found that the OPG values in the periodontitis I/II patients were higher than those in the periodontitis III/IV and healthy controls, although without any statistical significance between them.

We also calculated the RANK-L/OPG ratio and observed a higher RANK-L/OPG ratio in the patients with periodontitis III/IV compared with the patients with periodontitis I/II and healthy controls, albeit again without statistical significance (*p* = 0.055).

In Table 4, we summarize the characteristics of the salivary markers OPG, IL-1β, RANK-L and IL-17A in the patients classified based on the periodontitis progression rate into three categories, slow, moderate and rapid progression, indicating grade A, B and C, respectively, as defined by Tonetti et al., 2018 [8].

The ANOVA analysis showed that the patients with grade C periodontitis presented a significantly higher mean level of RANK-L [50.72 ± 20.10 pg/mL] compared with the grade B periodontitis [38.03 ± 10.89 pg/mL] and grade A periodontitis [35.28 ± 9.58 pg/mL] (*p* = 0.035) patients (Figure 1).

Similarly, we noticed that the grade C periodontitis patients presented mean IL-1β values higher than those of the grade B and grade A patients, albeit without reaching statistical significance. The mean IL-17A values did not show statistically significant differences between the different grades, although it stands out that the IL-17A levels were slightly increased in the grade C patients. While we did not find any statistically significant differences in any of the three periodontitis grades in terms of the OPG level, the latter showed a tendency to decrease with the disease progression, being higher in the grade A patients than in the grade C patients.

In the grade analysis, we also calculated the RANK-L/OPG ratio and observed a higher RANK-L/OPG ratio in the patients with grade C periodontitis in comparison with the grades B and A patients, although the different did not reach statistical significance (*p* = 0.428).

### 3.3. Analysis of OPG, IL-1β, RANK-L and IL-17A Levels in Saliva According to Clinical Criteria

The characteristics of the periodontal indices in the patients with grades A, B and C periodontitis are summarized in Table 5.

The grade C periodontitis patients showed significantly higher levels of total CAL (*p* = 0.008) than the grade B periodontitis patients. The mean values of PI percentage, BOP percentage and PPD indices did not show statistically significant differences between the different grades. However, it is noteworthy that the levels were higher in grade C periodontitis patients. The number of teeth was lower in the patients with grade C periodontitis, albeit without reaching statistical significance.

### 3.4. Correlation of Clinical Parameters and Grade of Periodontitis (Spearman’s Correlation)

When we correlated the immunomediators with the clinical parameters within the different grades of periodontitis, we found that in the grade A periodontitis patients, the total PPD was positively and statistically significantly correlated with the OPG (r = 0.873; *p* < 0.05), while the biomarker IL-1β correlated strongly and statistically significant positively with the PI percentage (r = 0.927; *p* < 0.01) and PPD (r = 0.873; *p* < 0.05) (Table 6).

In the patients with grade B periodontitis, there was only a moderate statistically significant positive correlation between the biomarker IL-1β and the PI percentage (r = 0.604; *p* < 0.05) and the BOP percentage (r = 0.572; *p* < 0.05).

In the patients with grade C periodontitis, there was a moderate to strong positive correlation between the IL-1β and PPD (r = 0.493; *p* < 0.05) and CAL (r = 0.743; *p* < 0.01), in addition to between the biomarker RANK-L and CAL (r = 0.443; *p* < 0.05) and between the biomarker IL-17A and the periodontal indices PPD (r = 0.542; *p* < 0.01) and CAL (r = 0.583; *p* < 0.01).

## 4. Discussion

This study is one of the first to jointly assess the levels of IL-1β, IL-17A, RANK-L and OPG in the saliva and relate them to the current classification of periodontal diseases [8], applying the severity and complexity (stage) as well as the risk of progression (grade) criteria to enable the identification of a biomarker or combined biomarkers that could discriminate different periodontitis statuses, facilitate an early non-clinical diagnosis, and allow for detection using a highly specific and sensitive multiplex assay.

The pathogenesis of periodontitis involves a complex network affected by elements of the microbiota biofilm and host tissues. This network engages in cellular signaling through pathogen molecular patterns and host recognition receptors, which activate a large population of cells, from periodontal tissues to infiltrating leucocytes, which trigger proinflammatory transcription factors, in particular enhancing cytokine secretion (e.g., IL1-β, IL-8) [24].

A significant number of reports demonstrated a marked increase in the IL-1β levels during the periodontal inflammatory process and evidenced its importance as a good predictive biomarker of periodontitis [24,25,26]. In this study, we analyzed the saliva immunomediators levels according to the new classification, using stage (I/II and III/IV) and grade (A, B and C disease progression) criteria. We found that the IL-1β levels were higher in the periodontitis III/IV patients in comparison with the periodontally healthy controls (*p* < 0.01) and with the periodontitis I/II patients (although not achieving statistical significancy). However, we also noticed that IL-1β was predominantly evident in the patients with grade C periodontitis (*p* = 0.768), indicating a good association with bone loss.

Reports from Arias-Bujanda et al. [27], who used a meta-analytical approach, have demonstrated that IL-1β and MMP8 are relevant biomarkers, with both showing clinically fair effectiveness for periodontitis diagnosis. In addition, salivary IL-1β has proven to distinguish systemically healthy patients with untreated periodontitis from periodontally healthy ones, although this discriminatory potential was reduced in smokers. [28].

Moreover, by means of Spearman’s correlation, we observed that the patients with grade C periodontitis exhibited a moderate to strong positive correlation between IL-1β and the clinical parameters PPD (r = 0.493; *p* < 0.05) and CAL (r = 0.743; *p* < 0.01).

Work by Reddahi et al. [29], whose study analyzed some saliva biomarkers in the function of periodontitis grades B and C, found higher IL-1β values in patients with grade C compared to patients with grade B (*p* < 0.001), as well as higher levels of IL-6, albeit not reaching statistical significance. They also observed that patients with grade C periodontitis had higher PPD scores than patients with grade B.

Taken together, our results agree with previous data [30,31,32] establishing an association between higher salivary IL-1β levels and the clinical status and higher presence of periodontitis in grade C periodontitis patients.

But can we associate salivary IL-1β with other immunomediators than can aggravate disease progression?

IL-1β is a cytokine produced by various cell types in the oral cavity, such as fibroblasts and endothelial cells, and by infiltrated leukocytes (mononuclear cells, macrophages and PMN). This cytokine is implicated in the process of inflammatory tissue destruction due to promoting osteoclast formation and bone reabsorption, where proteins such as RANK-L and OPG are implicated [33,34].

The advancement of the inflammatory process has been described as being implicated in the formation and activation of osteoclasts, the key cells in bone resorption. The triangle of RANKL/RANK/OPG regulates bone remodeling, with RANK expression in osteoclasts and other cells being activated by its ligand RANK-L, which is vital to osteoclasts formation, and contributing to bone destruction, and with OPG as a crucial protein, preventing the RANK-RANK-L bonding, thereby inhibiting osteoclast activation and promoting bone formation by osteoblasts [35,36,37].

Our analysis of the RANK-L levels permitted us to observe an association between the salivary RANK-L level and periodontitis III/IV stage (*p* < 0.001), as well as an enhanced RANK-L presence in the patients with grade C periodontitis, linking the RANK-L levels with rapid progression and bone loss. Moreover, in the grade C patients, a moderate positive correlation between the RANK-L presence and the clinical total CAL item was observed (* *p* < 0.05; ROC index Table 6). This attachment loss might be due to the excessive RANK-L production in the dental pocket, leading to continuous osteoclast activation and subsequent bone reabsorption and dental loss. In periodontitis stage III/IV and grade C disease status, IL-1β and RANK-L are both present and associated with the periodontitis stage severity and progression. Even if only RANK-L is statically different in grade C and not IL-1β, those cytokines are always higher is stage III/IV compared to stage I/II and healthy status.

In 2019, the study by Teodorescu et al. [38] in a population of patients affected by aggressive and chronic periodontitis observed significantly greater values of RANKL and the RANKL/OPG ratio in the saliva of the periodontitis patients in comparison with the healthy group, although no differences were observed between the two periodontitis stages, even if they were positively associated with bleeding and probing.

On the other hand, several studies demonstrated the RANK-L levels in GCF to be significantly higher in individuals with chronic periodontitis compared to periodontally healthy ones [39,40], as well as the OPG concentration to be more reduced in individuals with chronic periodontitis, although this difference was not always statistically significant [40,41]. This inconsistency between the data may result from several factors, such as the number and characteristics of the individuals, the collection and assay procedures, the specificity/sensibility of the immunoassay, and even the disease progression and activity over time.

Our data strongly suggest that the quantity of the IL-1β and RANK-L immunomediators in the saliva may be considered a readout and/or indirectly linked to the inflammatory condition and bone loss. As such, higher levels of IL-1β and RANK-L may contribute in sequence to an advanced stage and progression of periodontitis, in addition to being associated with PPD and CAL increases.

Taking into consideration the fact that stage III/IV is characterized by severity/extent and complexity, with the potential for an increase in the loss of dentition, and that grade C evidences a risk of rapid progression and estimates the potential impact of the disease, either systemically or regarding the predictability of standard periodontal therapy [8], these two immunomediators might work together as potential combined biomarkers in the salivary fluid and be of major help with the diagnosis of periodontitis.

As expected, our findings in terms of the clinical parameters clearly showed that the periodontitis III/IV patients had higher values for the plaque index (%PI), bleeding on probing (%BOP), total pocket depth (PPD) and clinical attachment loss (CAL) in comparison with the periodontally healthy controls (*p* < 0.001). These data are equally in agreement with other authors [15,42,43,44] and likely in relation to distinct gingival inflammatory situations [45].

Muhammad M. Majeed et al. [46] recently published a study in which increased levels of IL-1β were associated with high CAL and PPD parameters. In unstimulated whole saliva, the IL-1β levels were higher in periodontitis patients compared to healthy individuals, and severe cases presented higher values of IL-1β than moderate or initial periodontitis patients, even if applied to a low sample study.

IL-17A belongs to the IL-17 cytokine family (IL17A–IL17F), being mainly produced by the Th17 population, although it can also be secreted by TCD4+ cells, natural killer cells, neutrophils, eosinophils, mast cells and macrophages [47]. Its role in the process and regulation of inflammation is well known [48]. Our results concerning saliva IL-17A were less satisfactory due to being detected at considerably low levels. Despite this, they were higher in the higher stage and grade (grade C) group, albeit without achieving statistical significance. Yet, in the grade C periodontitis patients, IL17A presented a positive correlation with the periodontal indices, total PPD (r = 0.542; *p* < 0.01) and CAL (r = 0.583; *p* < 0.01).

Previous publications highlighted the role of IL17A in periodontitis [17]. The work of Tomás et al. (2017), who used a predictive accuracy model, identified IL17A as a good biomarker for distinguishing patients with chronic periodontitis from periodontally healthy individuals. Moreover, the predictive ability of this pro-inflammatory cytokine was increased by incorporating others, such as IL1β, IFN-γ and IL-10 [49].

We recognize several limitations of our study. The sample size could be larger, although it was sufficient to detect statistically significant differences between the groups. The inclusion of patients with gingivitis as another comparative group could highlight the very initial acute inflammatory process. In addition, creating a form of follow-up for the patients more regularly and for several years would be more reproduceable. We did not relate the immunomediators to the saliva microbiota. We did not use the Salimetrics Cotinine EIA cut points/levels to discriminate smokers from non-smokers. The literature refers to a relevant relationship between the microbial composition and cytokine pattern [50]. Therefore, exploring the saliva immunomediators and periodontal pathogens could have provided novel elements, either triggering/supporting disease progression or early diagnosis/prognosis. We found some discordance between studies in the literature, probably due to the different protocols for the collection, treatment and size of the sample studied and organization of data, but above all, due to the difficulty of establishing the dynamic states of activity and remission in chronic inflammatory diseases.

## 5. Conclusions

Our results are in agreement with the vast body of literature about the relevance of salivary IL-1β as a biomarker for chronic periodontitis. Here, it was present in the most severe and aggressive stage (II/IV) of the disease and correlated (Spearman’s correlation) with the pocket depth (PPD) and clinical attachment loss (CAL) in the grade C periodontitis status. It is highly relevant to note that salivary RANK-L was also detected in all the stages of periodontitis (I/II and III/IV), related to periodontitis Grade C and correlated with clinical attachment loss (CAL). Our results could not discriminate mild from severe periodontitis, but in addition to IL-1β, RANK-L could be taken as a combined diagnostic biomarker for periodontitis.

## Figures and Tables

**Figure 1 jcm-12-01003-f001:**
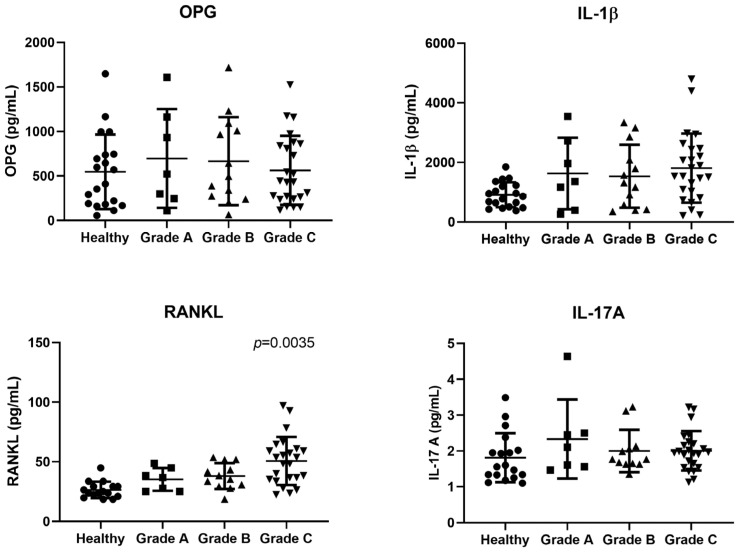
Scatter plots of IL-1β, OPG, IL17-A and RANK-L salivary levels (pg/mL) in the different grade groups. OPG: osteoprotegerin: IL-1β: interleukin -1β; RANK-L: receptor activator of nuclear factor kappa-B ligand; IL-17A: inteleukin- 17A; RANK-L/OPG: RANKL/OPG ratio; Horizontal lines represent M(SD) = mean (standard deviation). Statistical analysis with *p* value derived from the ANOVA one-way test and Bonferroni test (*p* = 0.035).

**Table 1 jcm-12-01003-t001:** Age, gender and smoking habit in healthy and periodontitis stage I/II and III/IV patients.

	Healthy Periodontium Subjects (n = 22)	Patients with Periodontal Disease I/II (n = 17)	Patients with Periodontal Disease III/IV (n = 29)	χ^2^	*p*
Age					
≤25 years	10 (90.9)	1 (9.1)	0 (0)	28.75	<0.001
≥26 years and ≤45 years	9 (42.9)	3 (14.3)	9 (42.9)		
≥46 years	3 (8.3)	13 (36.1)	20 (55.6)		
Gender					
Male	3 (13.6)	8 (47.1)	10 (34.5)	5.33	0.070
Female	19 (86.4)	9 (52.9)	19 (65.5)		
Smoking Habits					
Smoker	3 (21.4)	1 (7.1)	10 (71.4)	9.54	0.049
Ex-smoker	1 (12.5)	2 (25.0)	5 (62.5)		
Non-smoker	18 (39.1)	14 (30.4)	14 (30.4)		
N° cigarettes/day					
Less than 10 cigarettes	1 (33.3)	1 (100)	5 (50)	7.35	0.290
Between 10 and 20 cigarettes	0 (0.0)	0	1 (10)		
More than 20 cigarettes	2 (66.7)	0	4 (40)		

Note: Values indicate means (standard deviations) and number of subjects.

**Table 2 jcm-12-01003-t002:** Periodontal indices in periodontally healthy and periodontitis I/II and III/IV patients.

Periodontal Indices	Healthy (n = 22)	I/II (n = 17)	III/IV (n = 29)	Overall *p*	H *v*/*s* I/II *p*	H *v*/*s* III/IV *p*	I/II *v*/*s* III/IV *p*
PI%	19.27 ± 12.33	41.82 ± 18.13	65.21 ± 26.01	<0.001	0.003	<0.001	0.001
BOP%	6.14 ± 4.89	23.71 ± 14.79	47.05 ± 28.23	<0.001	0.026	<0.001	<0.001
PPD Total (mm)	1.65 ± 0.25	2.5 ± 0.69	3.68 ± 1.16	<0.001	0.008	<0.001	<0.001
CAL Total (mm)	1.67 ± 0.26	2.83 ± 0.86	4.30 ± 1.27	<0.001	0.001	<0.001	<0.001
Number of Teeth	28.18 ± 2.02	26.41 ± 3.79	23.21 ± 5.34	<0.001	0.570	<0.001	0.041

H: periodontally healthy patients; I/II: patients with periodontal disease I/II; III/IV: patients with periodontal disease III/IV; PI: plaque index; BOP: bleeding on probing; PPD: pocket depth; CAL: clinical attachment loss; M(SD) = mean (standard deviation); *p* = level of significance.

**Table 3 jcm-12-01003-t003:** OPG, IL-1β, RANK-L, and IL-17A salivary levels in periodontally healthy and periodontitis I/II–III/IV patients.

	Healthy	I/II	III/IV	*p*
OPG (pg/mL)	547.03 ± 418.12	701.60 ± 439.50	559.94 ± 441.68	0.489
IL-1β (pg/mL)	913.25 ± 418.20 *	1429.38 ± 1037.22	1866.25 ± 1152.15 *	0.006
RANK-L (pg/mL)	28.18 ± 9.68 **	37.10 ± 10.88 **	49.14 ± 19.55 **	<0.001
IL-17A (pg/mL)	1.82 ± 0.68	1.93 ± 0.50	2.03 ± 0.56	0.486
RANK-L/OPGratio	0.079 ± 0.048	0.093 ± 0.082	0.132 ± 0.082	0.055

H: periodontally healthy; I/II: patients with periodontal disease I/II; III/IV: patients with periodontal disease III/IV. OPG: osteoprotegerin: IL-1β: interleukin -1β; RANK-L: receptor activator of nuclear factor kappa-B ligand; IL-17A: inteleukin- 17A; RANK-L/OPG: RANKL/OPG ratio; Data summarized as mean and standard deviation; *p* value derived from the ANOVA one-way test and Bonferroni test. * significant difference from the periodontal disease III/IV and periodontally healthy patients (*p* = 0.004); ** significant difference from the periodontal disease III/IV group and periodontal disease I/II group (*p* = 0.045) and periodontal disease III/IV and periodontally healthy patients (*p* < 0.001).

**Table 4 jcm-12-01003-t004:** OPG, IL-1β, RANK-L, and IL-17A salivary levels in periodontitis grades A, B and C.

	Grade A Periodontitis (n = 7)	Grade B Periodontitis (n = 13)	Grade C Periodontitis (n = 26)	*p*
OPG (pg/mL)	696.65 ± 553.59	666.57 ± 494.14	562.66 ± 387.64	0.695
IL-1β (pg/mL)	1628.27 ± 1199.51	1533.91 ± 1058.89	1808.63 ± 1160.02	0.768
RANK-L (pg/mL)	35.28 ± 9.58	38.03 ± 10.89	50.72 ± 20.10	0.035
IL-17A (pg/mL)	1.95 ± 0.46	2.00 ± 0.59	2.01 ± 0.54	0.973
RANK-L/OPG ratio	0.098 ± 0.070	0.099 ± 0.092	0.133 ± 0.086	0.428

OPG: osteoprotegerin: IL-1β: interleukin -1β; RANK-L: receptor activator of nuclear factor kappa-B ligand; IL-17A: inteleukin- 17A; RANK-L/OPG: RANKL/OPG ratio; Data summarized as mean and standard deviation; *p* value derived from the ANOVA one-way test and Bonferroni test. M(SD) = mean (standard deviation); *p* = level of significance.

**Table 5 jcm-12-01003-t005:** Periodontal indices in patients with grade A, B and C periodontitis.

Periodontal Indices	Grade A Periodontitis (n = 7)	Grade B Periodontitis (n = 13)	Grade C Periodontitis (n = 26)	*p*
PI%	42.29 ± 16.79	53.69 ± 28.64	61.85 ± 25.50	0.186
BOP%	27.14 ± 19.63	34.46 ±26.11	43.44 ± 27.87	0.295
PPD total	2.93 ± 0.91	2.73 ± 0.92	3.58 ± 1.23	0.065
CAL total	3.24 ± 1.16	3.01 ± 0.95 *	4.27 ± 1.34 *	0.008
Number ofTeeth	26.14 ± 3.84	25.69 ± 4.91	23.26 ± 5.23	0.226

PI: plaque index; BOP: bleeding on probing; PPD: pocket depth; CAL: clinical attachment loss; Data summarized as mean and standard deviation; *p* value derived from the ANOVA one-way test and Bonferroni test; * significant difference from grade B periodontitis and grade C periodontitis (ANOVA *p* = 0.008 and Bonferroni test *p* = 0.012).

**Table 6 jcm-12-01003-t006:** Correlation between immunomediators and periodontal indices in patients with grade A, B and C periodontitis.

	PI%	BOP%	Total PPD	Total CAL
Periodontitis Grade A (n = 7)				
OPG	0.408	0.414	0.873 *	0.714
IL-1β	0.927 **	0.396	0.873 *	0.429
RANK-L	0.299	0.336	0.156	0.360
IL-17A	0.203	0.029	−0.290	−0.086
Periodontitis Grade B (n = 13)				
OPG	−0.214	−0.091	−0.120	−0.257
IL-1β	0.604 *	0.572 *	0.206	0.158
RANK-L	−0.175	−0.427	−0.487	−0.453
IL-17A	−0.091	−0.070	−0.365	−0.308
Periodontitis Grade C (n = 26)				
OPG	−0.158	−0.219	−0.021	0.208
IL-1β	0.127	0.176	0.493 *	0.743 **
RANK-L	−0.049	0.183	0.392	0.443 *
IL-17A	0.205	0.097	0.542 **	0.583 **

OPG: osteoprotegerin: IL-1β: interleukin -1β; RANK-L: receptor activator of nuclear factor kappa-B ligand; IL-17A: inteleukin- 17A; RANK-L/OPG: RANKL/OPG ratio; Analysis by Spearman’s correlation. * *p* < 0.05 ** *p* < 0.01.

## Data Availability

The data can be accessed by contacting the corresponding author.

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
