# Peer review of "Analysis of Salivary Levels of IL-1β, IL17A, OPG and RANK-L in Periodontitis Using the 2017 Classification of Periodontal Diseases—An Exploratory Observational Study"

_jcm, 2023, doi:10.3390/jcm12031003_

Round 1
Reviewer 1 Report
Authors aimed to assess the variation of salivary interleukin concentrations according to the stage and grade of periodontitis.
The material and methods are clearly described and the population included in the groups corresponds well to the definition of the stages and grades of periodontitis.
However, questions arise:
- were patients with autoimmune or autoinflammatory diseases included?
the periodontal health group is younger and the distribution of smokers/former smokers is different between the groups > this should be discussed.
- was salivary flow taken into account in the analyses?
The calculation of the RANKL/OPG ratio is to be calculated
In their analysis of the results, the authors do not really bring new scientific elements. The variation of markers in accordance with the state of periodontal health has already been described (although not according to the new classification).
Moreover, one may wonder
- if an analysis of several markers (among thse tested) could better distinguish patient profiles
- at what point would the results help to identify the stage or the grade? an increase of IL-1 will signify an advanced stage or an advanced grade? It seems to me that in the classification the idea of biomarkers was for the grade. One can indeed be in stage IV and have few teeth (so less pockets than a stage III).
If the authors wish to work on on biomarkers as an aid to non-clinical diagnosis, identifying gingivitis versus stage I/II versus stage IIII/IV periodontitis would be interesting.
Total PPD = mean attachment loss?
The discussion is too long
Author Response
Thank you very much for agreeing to review this article.
In relation to the questions raised, we have to say the following:
- were patients with autoimmune or autoinflammatory diseases included?
No, it did not. We have already included in the material and methods (line 102).
"medical history of diabetes mellitus or hepatic or renal disease, autoimmune and autoinflammatory diseases, or other serious medical conditions or transmittable diseases..."
- the periodontal health group is younger and the distribution of smokers/former smokers is different between the groups > this should be discussed.
Dear reviewer: Here, we justify the reasons for not including smokers/non-smokers results and discussion. However, if in the end you still consider it relevant, we will make the insertion according to your suggestion.
In well established in the literature than periodontitis is more severe in older subjects than is younger ones. Our sample is clinically classified and includes healthy subjects from age 18 to 25; 26-45 and older than 45, the frequency of these subjects in randomized. The distribution is accordance with other published works, were healthy group has a mean size lower than periodontitis groups.
The sample size depends on the convenience of acceptance by the patients, therefore it cannot be totally controlled. The sample of our healthy subjects is done at random and in accord with their acceptance. The sample of patients, as well, but subdivided in function of periodontal clinical parameters.
Other works have discribed that smokers had lower diagnostic threshols for IL 1ß in saliva than non-smokers. We analysed this two conditions (smokers/non-smokers) with our four immunomediators and we found no statistical diferences comparing RANK-L and IL17-A : smokers ›non-smokers and with OPG and IL1ß: non smokers › smokers. we did not included this data in the article, because we were not comparing smokers with non-smokers.
- was salivary flow taken into account in the analyses?
No, but the quantitative analysis using 4-plex kit Flow Assay by BioLegend was performed with the same volume of saliva (25uL), from a total volume of 1mL from all patients samples.
- The calculation of the RANKL/OPG ratio is to be calculated
We have calculated the RANKL/OPG ratio and included it in tables 3 and 4. The results were described in lines 232-234 and 257-259).
- In their analysis of the results, the authors do not really bring new scientific elements. The variation of markers in accordance with the state of periodontal health has already been described (although not according to the new classification).
The originality is undoubtedly the selection of patients according to the new classification of periodontal diseases. No study until now, has simultaneously analysed these four biomarkers distributed according to the new classification; more another advantage was that the immunomediators were evaluated together, on the same sample, using a more sensitive technique.
- if an analysis of several markers (among thse tested) could better distinguish patient profiles
We think so, because two of this cytokines (IL-1beta, IL-17A) are representative of the inflammatory process giving rise to distinguishing stage periodontitis. The other parameters relevant to bone remodeling (OPG,RANKL) would allow to distinguish grades.
- - at what point would the results help to identify the stage or the grade? an increase of IL-1 will signify an advanced stage or an advanced grade? It seems to me that in the classification the idea of biomarkers was for the grade. One can indeed be in stage IV and have few teeth (so less pockets than a stage III).
In this study, we observed that IL-1ß level has a better association with stage than with grade (Table 3 versus Table 4) In the case of RANK-L, we observed a link with both, stage and grade. Being a clinical study, conditions cannot be changed. By these results, we could propose that RANK-L is more appropriate for the differentiation in grades. As this study is the first to use the new classification, I accept that further studies will have to be performed for a final definition. Nevertheless, combined IL-1ß and RANK-L would permit a better stage/grade identification.
- If the authors wish to work on on biomarkers as an aid to non-clinical diagnosis, identifying gingivitis versus stage I/II versus stage IIII/IV periodontitis would be interesting.
Thank you very much for your suggestion. It was mentioned in the limitations (discussion); we will take this into account in future studies.
- Total PPD = mean attachment loss?
No, PPD is Pocket deep, and is defined in M&M (line 110). CAL is clinical attachment loss (line 114).
- The discussion is too long
We took out paragraph in discussion (line 411 to 419).
Reviewer 2 Report
The Authors have explored the associations of a small group of biomarkers across patients with different diagnoses per the new classification of periodontal diseases. The study has merits, but some concerns are presented below.
· Your study design is not exactly that of cross-sectional studies since those studies recruit a large sample of the population to investigate associations. I would rephrase the study type to exploratory observational study. Exploratory because the problem is not clearly defined and observational because there is no intervention.
· How do you account for the sample size?
· How do you account for the choice of this specific group of biomarkers? What is your rationale?
· Please use neutral language for results and express opinions with adjective in discussion.
· Please provide references for your plaque index and bleeding score methods.
· Smoking can have substantial impact on systemic inflammation and potentially salivary biomarkers. Please add in the limitations section that you could not determine cotinine levels which is the gold standard to assess smoking habits.
· You mention throughout the text that certain biomarkers were expresses. Please not that expression refers to a genetic process which may or may not be reflected in the determination of in some biological fluid. Please do not use expression and biomarker determination interchangeably.
· There are some typos and grammar errors please send for language edits.
Author Response
Thank you very much for agreeing to review this article.
In relation to the questions raised, we have to say the following:
- Your study design is not exactly that of cross-sectional studies since those studies recruit a large sample of the population to investigate associations. I would rephrase the study type to exploratory observational study. Exploratory because the problem is not clearly defined and observational because there is no intervention.
Thank you very much for your sugestion. We changed the title was you sugested: "Analysis of salivary levels of IL-1β, IL17A, OPG and RANK-L in periodontitis using the 2017 classification of periodontal diseases – an exploratory observational study."
- How do you account for the sample size?
The sample size and, since it is a convenience sample, it was not possible to control it, because it depend on whether or not the patients accepted to participate in the study.
- How do you account for the choice of this specific group of biomarkers? What is your rationale?
The criterion is rational. Two are the main inflammatory cytokines (IL-1beta, IL-17A) and the other parameters relevant to bone remodeling (OPG/RANKL). These aspects are described in the introduction separately for each one and supported by the bibliography. These cytokines are therefore widely used to explore the inflammatory process. In this study, the advantage was that they were evaluated together, on the same saliva sample, using a more sensitive technique flow assay .
- Please use neutral language for results and express opinions with adjective in discussion.
We analysed again the article and made some modifications.
- Please provide references for your plaque index and bleeding score methods.
We introduced in the M&M in line 114 (references 19 and 20).
- Smoking can have substantial impact on systemic inflammation and potentially salivary biomarkers. Please add in the limitations section that you could not determine cotinine levels which is the gold standard to assess smoking habits.
In discussion, it was introduced in lines 422-423. "We did not use the Salimetrics Cotinine EIA cut points/levels, to discriminate smokers from non-smokers."
- You mention throughout the text that certain biomarkers were expresses. Please not that expression refers to a genetic process which may or may not be reflected in the determination of in some biological fluid. Please do not use expression and biomarker determination interchangeably.
We took out the words expressed ou expression in discussion ( lines 322 and 354).
- There are some typos and grammar errors please send for language edits.
Dear refereer we regret the spelling or language errors, however, we reread and made some changes.